# Effects of Long-Term Nitrogen Fertilization on the Formation of Metabolites Related to Tea Quality in Subtropical China

**DOI:** 10.3390/metabo11030146

**Published:** 2021-03-02

**Authors:** Yuzhen Chen, Feng Wang, Zhidan Wu, Fuying Jiang, Wenquan Yu, Jie Yang, Jiaming Chen, Guotai Jian, Zhiming You, Lanting Zeng

**Affiliations:** 1Tea Research Institute, Fujian Academy of Agricultural Sciences, No. 104 Pudang Road, Xindian Town, Jin’an District, Fuzhou 350012, China; taotaoyuzhen@163.com (Y.C.); 82458lin@163.com (F.W.); fjanxiwzd@163.com (Z.W.); fuyingjiang93@163.com (F.J.); 2National Agricultural Experimental Station for Soil Quality, No. 1 Hutouyang Road, Shekou Town, Fu’an 355015, China; 3Fujian Academy of Agricultural Sciences, No. 247 Wusi Road, Gulou District, Fuzhou 350013, China; ywq1972@163.com; 4Key Laboratory of South China Agricultural Plant Molecular Analysis and Genetic Improvement & Guangdong Provincial Key Laboratory of Applied Botany, South China Botanical Garden, Chinese Academy of Sciences, No. 723 Xingke Road, Tianhe District, Guangzhou 510650, China; yangjie0727@163.com (J.Y.); chenjiaming@scbg.ac.cn (J.C.); jiangt@scbg.ac.cn (G.J.)

**Keywords:** long-term nitrogen fertilization, *Camellia sinensis*, tea quality, chlorophyll, catechin, l-theanine, aroma compound

## Abstract

As a main agronomic intervention in tea cultivation, nitrogen (N) application is useful to improve tea yield and quality. However, the effects of N application on the formation of tea quality-related metabolites have not been fully studied, especially in long-term field trials. In this study, a 10-year field experiment was conducted to investigate the effect of long-term N application treatments on tea quality-related metabolites, their precursors, and related gene expression. Long-term N application up-regulated the expression of key genes for chlorophyll synthesis and promoted its synthesis, thus increasing tea yield. It also significantly increased the contents of total free amino acids, especially l-theanine, in fresh tea leaves, while decreasing the catechin content, which is conducive to enhancing tea liquor freshness. However, long-term N application significantly reduced the contents of benzyl alcohol and 2-phenylethanol in fresh tea leaves, and also reduced (*E*)-nerolidol and indole in withered leaves, which were not conducive to the formation of floral and fruity aroma compounds. In general, an appropriate amount of N fertilizer (225 kg/hm^2^) balanced tea yield and quality. These results not only provide essential information on how N application affects tea quality, but also provide detailed experimental data for field fertilization.

## 1. Introduction

Tea plant is one of the world’s most important cash crops and tea products are among the most popular non-alcoholic beverages and health foods. The widespread consumption of tea is attributed to its color, aroma, and benefits to human health [1]. The characteristic properties of tea are determined by the abundant metabolites in tea, such as chlorophyll and some pigments related to tea color [2], catechins, amino acids, caffeine, aroma compounds, and other metabolites related to taste, aroma, and function [3,4,5,6,7]. Therefore, it is important to study the formation and regulation of metabolites related to tea quality. In addition to differences among tea cultivars, the formation of metabolites related to tea quality is determined by growth conditions, agronomic practices in the preharvest stage, and postharvest processing technology [8,9,10,11]. There have been many studies on the influence of environmental factors and processing technology on the formation of tea quality-related metabolites. In recent years, numerous studies have investigated the effect of agronomic interventions on tea quality-related metabolites. Fertilizer application is an important agronomic factor in tea production. The main reason is that tea trees are a perennial leaf cash crop, most are planted in poor soils such as hillside lands, and tea shoots are subject to picking and pruning several times a year, processes that take away a lot of nutrients from the tea plant. As a result, tea plants have a high demand for fertilizers, especially nitrogen (N) [12].

N application can improve the nutrient supply from the soil, thereby preserving the normal physiological metabolism of tea plants, which is very important for maintaining tea quality and yield [13]. According to official statistics, high N application cultivation is universal in tea plantations in China; the national average application is 491 kg/hm^2^ N per year [14]. However, the utilization rate of N fertilizer is only about 30%, while the rest is lost through ammonia volatilization, N_2_O emission, surface leaching and underground runoff, resulting in fertilizer waste and environmental pollution [15,16,17]. It has been shown that reasonable application of N fertilizer (200–350 kg/hm^2^) can significantly increase tea yield and also affect quality-related components of fresh tea leaves, such as polyphenols, amino acids, caffeine and aroma compounds [18,19,20,21]. For example, appropriate N fertilizer application can increase the free amino acids and chlorophyll in fresh tea leaves, reduce the polyphenols and phenol–ammonia ratio [20], and also maintain the balance between lipid metabolism and aroma compound formation. Excessive N fertilizer application can inhibit the synthesis of soluble sugars and polyphenols in fresh tea leaves, increase the contents of precursors for grass-like aroma compounds and arginine, which tastes bitter and astringent, and thus can detract from tea taste and aroma [20]. Although numerous studies have shown that N fertilization can affect quality-related metabolites of tea leaves, the underlying mechanism is not clear. In addition, recent studies have mainly been conducted in artificial climate chambers with hydroponic or pot cultivation, while there is a lack of long-term field experiments to study the realistic response to N fertilizer. For perennial tea plants, a large proportion of N is stored in the roots, stems, and mature leaves, and remobilized N in each organ is the main source for absorption and accumulation in the young shoots [22]. Therefore, the effect of fertilizer is not obvious in short-term field experiments because of the large amounts of nutrients stored in tea plants, especially in mature plants (10–30 years). In addition, the N fertilizer use efficiency of tea plants is affected by climate, soil properties (texture, forms of available N, and pH), N-demand characteristics of different cultivars, picking season, and fertilization time. Therefore, it is necessary to comprehensively investigate the effect of long-term N application on the formation of tea quality-related metabolites in field experiments.

In this study, a long-term experiment (10 years of N fertilization) was conducted to investigate the effect of different N application treatments on chlorophyll, catechin, free amino acids, caffeine and aroma compounds, their precursors and characteristic gene expression related to those metabolites. This study provides the comprehensive profile of effect on tea quality-related metabolites involved in color, aroma, and taste, induced by long-term N application. On the one hand, the results of this study will help us to understand the intrinsic mechanism by which N application affects tea quality; on the other hand, the study provides reference data from field experiments for setting the appropriate range of N application for optimum tea yield and quality in commercial production.

## 2. Results

### 2.1. Effects of Long-Term Nitrogen Application on Tea Yield

Budding density and 100-bud weight are important indicators for the growth and yield of tea plants. For spring tea, compared with the treatment without N fertilizer (N0), three long-term N application treatments significantly increased the budding density and 100-bud weight by 48.89–114.44% and 58.58–148.57%, respectively (Table 1). However, these two indexes showed no significant difference between N2 and N3 treatments. For autumn tea, three long-term N treatments increased the budding density and 100-bud weight by 17.07–25.61% and 11.91–33.33% (Table 2). In addition, for budding density, there was no significant difference among the three N application treatments, but the 100-bud weight of N3 treatment was significantly higher than that of the other two N application treatments. Long-term N application increased the yield of spring and autumn tea by 137.79–430.20% and 33.43–67.49%, respectively, but there was no significant difference between N2 and N3 treatments (Table 1 and Table 2). Overall, the results showed that long-term N application evidently promoted tea yield, but it is just in a certain range. Therefore, appropriate N application is significant to balance the relationship between fertilizer input and yield increase in a tea garden.

### 2.2. Effects of Long-Term Nitrogen Application on Chlorophyll Formation in Tea Leaves

In the process of plant N metabolism, more than half of N is usually allocated to chloroplasts to participate in photosynthesis [23]. The content of photosynthetic pigments determines the intensity of photosynthesis. Figure 1A shows that, compared with N0, long-term N application increased the chlorophyll and carotenoid content in tea leaves, among which chlorophyll b increased with increasing rate of N applied, and chlorophyll a, total chlorophyll and a/b ratio were highest in N2 treatment. Compared with N0, chlorophyll a, chlorophyll b, and total chlorophyll contents in N2 and N3 treatments were significantly higher, but there was no significant difference among the three N application treatments.

The expression of characteristic genes involved in chlorophyll synthesis and degradation was investigated [24]. As shown in Figure 1B, after long-term N application, the expression of four characteristic genes for chlorophyll synthesis, including *glutamyl-tRNA reductases* (*CsHEMAs*) and *protochlorophyllide oxidoreductase* (*CsPORs*), tended to increase at first and then decrease with increasing N application, with the highest gene expression in the medium N treatment, 225 kg/hm^2^ (N2). Statistical analysis showed that the expression of *CsHEMA1*, *CsHEMA2,* and *CsPOR1* in fresh tea leaves under N2 treatment was significantly higher than N0 and N1 treatments, and the expression of four chlorophyll synthesis genes was not significantly different between N2 and N3 treatments. The expression of chlorophyll degradation gene (*pheophorbide a oxygenase*, *CsPAO*) was not significantly different among the four N application treatments (Figure 1C). In conclusion, long-term N application promoted chlorophyll content in tea leaves, probably as a result of increased expression of key genes for chlorophyll synthesis, and the chlorophyll content was highest in N2 treatment, indicating that moderate N application was the most effective in promoting photosynthesis in tea leaves.

### 2.3. Effects of Long-Term Nitrogen Application on the Formation of Free Amino Acids in Tea Leaves

Determination on free amino acids in tea samples showed that the main components were l-theanine, l-glutamic acid, l-glutamine, serine, threonine, alanine, valine, asparagine, phospholysine, and phosphoserine, among which l-theanine accounted for around 75% (Table 3). The contents of total free amino acids and main components in fresh leaves increased significantly after long-term N application, especially in the medium and high N application treatments, i.e., 225 kg/hm^2^ and 450 kg/hm^2^. However, the contents of phosphoserine, glycine, tryptophan, and taurine in fresh tea leaves were not significantly different among the four N application treatments.

To explore how N fertilizer affects L-theanine formation in fresh tea leaves, genes for L-theanine synthesis, including *L-theanine synthase* (*CsTS1*) and *L-glutamine synthases* (*CsGSs*), and degradation (*pyridoxal 5′-phosphate synthase subunit*, *CsPDX*) were selected for quantitative real-time PCR (qRT-PCR) [25,26]. As shown in Figure 2, the expression of *CsTS1*, *CsGS1.1*, and *CsGS1.3* was lower in the N fertilizer treatments, and the expression of *CsGS2* and *CsGS1.2* was not significantly different among the four treatments. Furthermore, the expression of *CsPDX2.1* in fresh tea leaves in N1 treatment was significantly lower than that in N0 treatment, but there was no significant difference among N0, N2, and N3 treatments (Figure 2). In all, long-term N application had no effect on gene expression for l-theanine synthesis and degradation. Therefore, it was speculated that l-theanine increase might be the increase in content of l-glutamic acid, as one of the substrates for l-theanine synthesis.

### 2.4. Effects of Long-Term Nitrogen Application on Catechin and Caffeine Formation in Fresh Tea Leaves

The effect of long-term N application on the formation of catechin and caffeine, which are the main bitter and astringent factors in tea liquors, was investigated. Figure 3A shows that long-term N application significantly affected the formation of catechins in fresh tea leaves, but its effects on the various components were different. With increasing N application, the contents of total catechins and components, including epigallocatechin gallate (EGCG), epicatechin gallate (ECG), gallocatechin (GC), and catechin (C), decreased gradually. However, the response of catechin gallate (CG), epicatechin (EC), and epigallocatechin (EGC) in fresh tea leaves was not sensitive to long-term N application. Long-term N application significantly reduced the phenol–ammonia ratio by 51.11–78.49% (Figure 3B). The reason might be that long-term N application not only increased the content of amino acids, but also reduce the content of polyphenols, thus reducing the phenol–ammonia ratio in tea leaves, which is conducive to the formation of tea liquor freshness. In addition, there was no significant difference in caffeine content among the four N application treatments (Figure 3C), indicating that long-term N application had few effects on caffeine formation in fresh tea leaves.

### 2.5. Effect of Long-Term Nitrogen Application on the Formation of Aroma Compounds in Tea Leaves

The effects of long-term N application on the formation of aroma compounds in fresh tea leaves collected in spring were significantly different among different aroma kinds (Appendix A). Long-term N application had no significant effect on the three tea key aroma compounds, including linalool, (*E*)-nerolidol, and indole. However, the N application reduced benzyl alcohol and 2-phenylethanol derived from the l-phenylalanine metabolism pathway in fresh tea leaves (Figure 4 and Figure 5A). Furthermore, the gene expression of *phenylalanine lyase* (*CsPAL*) and *aromatic amino acid aminotransferase* (*CsAAAT*) in fresh tea leaves was significantly lower under medium and high N fertilization (i.e., N2 and N3), compared with N0 (Figure 5B). The results indicated that the decrease in aroma compounds derived from the l-phenylalanine pathway caused by long-term N application might be related to down-regulated gene expression.

The withering stage, which is used in the production of six tea types in China, plays an important role in the formation of metabolites, in particular aroma compounds [27]. Therefore, in this study, fresh leaves of autumn tea were withered for 2 h and 4 h, with 0 h after picking taken as the control. Among four different N application treatments, the content of indole and (*E*)-nerolidol in fresh tea leaves both increased progressively with withering (Figure 6). However, after withering for 2 h and 4 h, the contents of these two compounds in fresh tea leaves of N2 and N3 treatments were significantly lower than those in N0 treatment, which indicated that the long-term N application was not beneficial to the formation of floral aroma compounds during the withering process. Linalool oxide I and linalool oxide II in fresh tea leaves of N2 and N3 treatments both increased during the withering time, and both these aroma compounds were significantly higher than in N0 treatment. With extended withering time, the expression of most structural genes increased significantly, except for *myelocytomatosis proteins* (*CsMYC2s*, key transcription factors in jasmonic acid signaling) (Figure 7). At withering for 0 h, the expression of *lipoxygenase* (*CsLOX1*, responsible for jasmine lactone synthesis) and *CsMYC2s* was also lower after long-term N application. The expression of *(E)-nerolidol synthase* (*CsNES*) and *tryptophan synthase α-subunit 2* (*CsTSB2*), responsible for (*E*)-nerolidol and indole synthesis [28,29,30], was lower in higher N application treatments after 4 h withering (Figure 7). However, there was no significant difference among the four N application treatments in the expression of *tryptophan synthase β-subunit* (*CsTSA*, interacting with *CsTSB2* for indole synthesis) after the different withering times.

## 3. Discussion

### 3.1. Effect of Long-Term Nitrogen Application on Chlorophyll Synthesis in Fresh Tea Leaves

N is the most important nutrient element for the growth and quality formation in tea plants, and it plays an important role in the nutritional and physiological functions of plant [31,32,33]. In tea cultivation, N application can promote the nutrient accumulation in the leaves, as well as the germination and growth of plants, thus greatly improving the yield. In this study, it was found that moderate N application was sufficient for the growth of tea plants (Table 1 and Table 2), which was consistent with previous studies [20,33,34,35]. 

Chlorophyll is a centrally important pigment in the photosynthetic pathway within the chloroplasts, and plays a crucial role in light energy capture and transfer. In addition, it is the material basis for the color of tea leaf and liquor, and the foundation for aroma compound formation [31,36]. Therefore, its content and variation are crucial to tea quality. The biosynthesis of chlorophyll is an enzyme-catalyzed process encoded by a series of genes [24]. Many environmental factors, such as light (intensity and quality), temperature, moisture, soil mineral elements (especially N and Mg), can affect enzyme activity and gene expression, thus affecting chlorophyll synthesis [37,38,39]. N is an essential nutrient for chlorophyll synthesis, and its availability directly affects the chlorophyll content. N application within a certain range can increase the contents of chlorophyll a and b in fresh tea leaves (Figure 1A), which was consistent with the previous studies [35]. *CsHEMA* encodes the enzyme responsible for the conversion of l-glutamyl-tRNA into chlorophyll, *CsPOR* encodes the enzyme that converts protochlorophyllide to chlorophyllide, while *CsPAO* is the key enzyme in the chlorophyll degradation pathway. All enzymes for chlorophyll biosynthesis and degradation have been identified in model plants [24,40]. Most of the studies focused on chlorophyll synthesis and degradation in relation to tea leaf color [41,42]. The changes in expression of genes involved in chlorophyll synthesis and degradation (Figure 1B,C) indicated that the increase in chlorophyll content in fresh tea leaves under long-term N application might be due to the activation of genes involved in its synthesis [43,44]. The accumulation of chlorophyll in plants is a dynamic process, and the balance of its synthesis and degradation is complex. However, few researchers have studied the regulation of the relevant genes in tea plants, and this topic needs to be further explored.

### 3.2. Effect of Long-Term Nitrogen Application in Promoting Amino Acid Synthesis

Amino acids, which are products of N metabolism in tea plants, are the primary metabolites determining the freshness of tea taste [4,31]. In tea plants, they are also the material basis for protein synthesis, precursors for nucleic acids, chlorophyll, phytohormones, and the main transport form of reduced N [45]. Apart from the precursors, other factors, such as soil type, humidity, the number of sunny day, and plant elicitation, may also affect the formation of secondary metabolites [46]. The synthesis and transport of amino acids depend on N availability, and both ammonium and nitrate fertilizer application can increase the contents of amino acids in tea leaves, especially ammonium. Previous research showed that, with increasing application of N within a limited range, although l-theanine, l-glutamic acid, and l-glutamine increased, high arginine content might lead to the reduction in taste quality [20]. The contents of aspartic acid, l-glutamic acid, and l-theanine, increased significantly with ammonium treatment, and the reason might be that the ammonium promoted root growth and inhibited root browning [47]. In our study, the increase in total free amino acids and several major amino acid components (Table 3), especially in the medium and high N treatments, i.e., N2 and N3, has also been reported in other plants [48].

The synthesis of l-theanine in tea plants has been a hot topic in the study of N assimilation and absorption. To date, several important enzymes involved in the l-theanine metabolism pathway have been isolated and identified, such as l-theanine synthase, l-glutamine synthase, and l-theanine hydrolase [25,49,50]. However, the functions of related genes have not yet been proved definitively in transgenic tea plants. In recent years, with publication of tea genome [51,52,53], the functional characterization of these genes has begun to become clear. The cytosol was the main site of l-theanine biosynthesis in tea root tissue, and cytosol-expressed *CsTS1* was shown to encode l-theanine synthase. In tea shoot tissue, l-theanine biosynthesis occurred mainly in the cytosol and chloroplasts, and *CsGS1.1* and *CsGS2* most likely encode the key l-theanine synthases. *CsPDX2.1* was a key enzyme that catalyzes l-theanine hydrolysis into ethylamine and glutamate, and the hydrolysis occurred mainly in the mitochondria and peroxisome [25,49]. In the study, the trend in expression of genes involved in l-theanine metabolism with increasing N application was not consistent with its accumulation in the tea leaves (Table 3 and Figure 2), which might be the result of differences in l-theanine synthesis among different organs and tea cultivars [49,54]. l-Theanine is mainly synthesized in the roots of tea plants and then transported to the stems and leaves. The transfer and utilization rates are distinct among different tea cultivars, so the related gene expression might not fully reflect l-theanine accumulation in tea plants. From the above results, we suggest that long-term N application provides more carbon skeletons and energy for ammonia assimilation into amino acids, and increases the N allocation to free amino acids, so that amino acids in tea leaves accumulate rapidly. However, the relationship between l-theanine synthesis in the roots and decomposition in the leaves of tea plants remains to be further explored.

### 3.3. Effect of Long-Term Nitrogen Application in Decreasing Metabolites Derived from l-Phenylpropanoid Metabolism

In tea plants, l-phenylalanine is a major amino acid, and its metabolic pathway is important for synthesis of many secondary metabolites. Polyphenols (e.g., catechins, flavonoids, lignin, anthocyanins and coumarins), and aromatic aroma compounds (e.g., benzyl alcohol, 2-phenylethanol, phenylacetaldehyde and benzaldehyde), derived from l-phenylalanine, are closely related to tea quality and stress resistance in tea plants [28,55]. During the growth of tea plants, the light conditions, temperature, drought, pathogens, and insect feeding can significantly increase the content of catechins, lignin, and volatile phenylpropanoids and benzenoids [6,56]. During the tea manufacturing process, withering, rolling, and fermentation can significantly increase aromatic aroma compounds in tea leaves [5,8,30]. However, there are few reports on the effects of long-term N fertilization on the l-phenylpropanoid pathway in tea leaves. Many of the carbon-containing compounds produced by photosynthesis are used to synthesize protein under conditions of N insufficiency or excess, which restricts the conversion of some sugars to polyphenols and results in an overall decrease in these compounds [20]. A short-term field experiment showed that optimized N fertilization significantly reduced the phenol–ammonia ratio in tea leaves [16], and the same phenomena also occurred in the long-term N application (Figure 3B). In addition, long-term N application decreased the content of benzyl alcohol and 2-phenylethanol (Figure 5A), affecting the formation of aroma compounds during the withering process of tea. 

PAL is the rate-limiting enzyme in plants connecting primary metabolism with secondary metabolism of phenylpropanoids. At present, six *PAL* genes in tea plant have been reported [57]. Phenolic compounds are synthesized by the phenylpropanoid pathway initiated by PAL [58]. AAATs catalyze l-phenylalanine into phenylpyruvic acid in tea plants, and *CsAAAT2* was proposed as the key one [28]. The expression of *CsPAL* and *CsAAAT2* under long-term N fertilization was significantly lower than that with no N fertilization (Figure 5B), consistent with previous studies on the effect of N application on tea plants using RNA-Seq [13,19,59]. From the above results, we hypothesized that, on the one hand, as a result of long-term N application, the accumulation of chlorophyll in tea leaves increased and more carbon-containing compounds were used to synthesize amino acids, while the concentration of substrate for the synthesis of polyphenols was relatively reduced, which led to the reduction in the content of catechins; on the other hand, the down-regulation of *CsPAL* and *CsAAAT2* involved in the l-phenylalanine pathway might be the main factor leading to the decrease of these metabolites in this pathway. Since the regulation of the l-phenylalanine pathway in tea plants has not been fully elucidated, further studies on the effects of long-term N fertilization on this pathway are essential.

### 3.4. Long-Term Nitrogen Application Might Affect the Formation of Floral Aroma Compounds in Tea Leaves during the Withering Process

The withering stage can affect the formation of tea quality-related metabolites [7,27,60,61]. Withering, involving drought, wounding, heat, and UV radiation greatly affect the formation of aroma compounds in tea leaves [6,8,30]. For example, contents of characteristic floral and fruity aroma compounds (e.g., indole, (*E*)-nerolidol and jasmine lactone) were significantly enhanced during the withering stage of oolong tea manufacture, as a result of wounding stress [29,30,62]. Indole at a low concentration is floral [63], and it acts as an important aroma contributor of oolong tea [64,65]. Our previous study showed that long-term N fertilization caused apparent changes in free amino acids, catechins, and aroma compounds in tea leaves, but whether the differences in these metabolites affected the formation of aroma compounds in tea leaves during the withering stage of tea manufacture was unclear. Continuous withering can induce accumulation of characteristic aroma compounds, such as indole and (*E*)-nerolidol, and our results were similar to previous studies on oolong tea manufacture [6,66,67,68]. Some genes involved in the final step of (*E*)-nerolidol and indole biosynthesis have been isolated and identified [30,62]. In the present study, only *CsTSB2* was significantly up-regulated by prolonged withering in all of the samples, while *CsTSA* was not significantly affected (Figure 7), which was consistent with the previous study [30]. In the study, continuous withering could activate the expression of *CsNES* and *CsTSB* (Figure 7), which are well consistent with previous research results [30,68]. In addition, it could be speculated that long-term N application decreased metabolites derived from the l-phenylpropanoid metabolic pathway, and so the content of intermediates in the synthesis of volatile phenylpropanoids and benzenoids was lower (Figure 5A).

Many previous studies have shown that wounding stress plays an important role in the indole formation during postharvest manufacturing [30,64,66]. Furthermore, indole may also be formed as a result of degradation, possibly by microorganisms. However, compared with dark tea, there are relatively few studies on microbial decomposition to produce quality-related metabolites, especially aroma compounds, during oolong tea manufacturing. Systematical investigations on the occurrence of microorganisms in the oolong tea manufacturing, and the specific microbial community composition, need to be carried out in the future. Only based on these investigations, the question of whether microorganisms are involved in the aroma formation during oolong tea manufacturing might be answered. During the withering process of oolong tea, the aroma compounds are in a dynamic state, involving multiple synthetic and degradation pathways, and microorganisms may also participate in their formation. In future studies, we will use an isotope tracer method to quantitatively characterize the formation and degradation of aroma compounds during the withering process and overcome the limitations of current research.

## 4. Materials and Methods

### 4.1. Experimental Design and Different Fertilization Treatments

The field experiment was set up in the base (119°57′ E; 27°22′ N; approximately 91 m above sea level) of the Tea Research Institute, Fujian province, China. Tea plants were grown in an experimental field, ensuring that nutrients would not flow and cross-contaminate among the experimental plots. The size of pool was 2 × 0.9 × 0.9 m (Figure 8). The soil is derived from weathered granite, and belongs to humic acrisols (World Reference Base for Soil Resource), which has a sandy loam texture. The basic properties of the surface soil (0–20 cm) were: pH 4.85, soil organic carbon 3.7 g/kg, total N 0.3 g/kg, available N 26.4 mg/kg, available P 4.8 mg/kg, available K 60.3 g/kg, C/N ratio 12.4, cation exchange capacity 4.61 cmol/kg and bulk density 1.00 g/cm^3^. The tea cultivar is “You 4” and five tea trees were planted in each pool with about 30 cm apart between each plant. The 3-year-old tea tree was planted in March 2010.

For the tea garden in Fujian province, China, the average amount of chemical fertilizers is 266 kg/hm^2^ N, 186 kg/hm^2^ P_2_O_5_, and 193 kg/hm^2^ K_2_O [14]. Therefore, four fertilization treatments were designed in this experiment, including N0 (0 N kg/hm^2^), N1 (112.5 N kg/hm^2^), N2 (225 N kg/hm^2^), and N3 (450 N kg/hm^2^), and each treatment was repeated four times. Urea, superphosphate (150 kg/hm^2^ P_2_O_5_), and potassium sulfate (150 kg/hm^2^ K_2_O) were applied to the tea garden soil as nitrogen, phosphorus, and potassium fertilizers, respectively. The nitrogen and phosphate fertilizer were applied in split three applications (30:30:40), and the potassium fertilizer was only used as basal manure in winter. The time of first fertilization began in the winter of 2011. Fertilizer application rate remained the same each year. The management of experimental plot was consistent with that of a conventional tea garden.

### 4.2. Tea Leaves Sampling and Analysis

In general, the fresh tea leaves picked in spring are usually manufactured into green tea, and the ones picked in autumn are mainly manufactured into black tea and oolong tea. As the spring tea fresh leaves are processed by directly panning into green tea and the autumn tea fresh tea leaves are manufactured into black and oolong tea by a long withering stage, we explored the relevant indicators in the fresh leaves picked in spring and the variations of aroma compounds in tea leaves picked in autumn during the withering process [69,70]. Therefore, the fresh tea leaves in spring were directly collected, and the ones in autumn were collected after withering treatment. All tea leaves were plucked after 10 years of applying nitrogen in the above-mentioned quantities. The details about the sampling of tea leaves in spring are shown below. On 17 April 2020, one bud and two leaves from four N application treatments (N0, N1, N2, N3) were picked, then immediately frozen with liquid nitrogen, and stored at −80 °C for subsequent determination. Each treatment was repeated four times, with a total of 16 samples. The details about the sampling of tea leaves in autumn are shown below. On October 8 2020, one bud and two leaves from four N application treatments (N0, N1, N2, N3) were picked, and stacked for 0 h, 2 h, and 4 h. All the treatments were repeated three times with a total of 36 samples. After treatment, all the samples were frozen with liquid nitrogen, and stored at −80 °C for subsequent determination.

### 4.3. Determination of Chlorophyll and Carotenoids

The method for determination of chlorophyll and carotenoids in tea samples was slightly modified from the previous research methods [71,72]. A total of 100 mg of tea leaves (fresh weight, finely powdered) was extracted with 5 mL 80% acetone under dark condition at 25 °C for 24 h. The extract was centrifuged for 5 min (4 °C, 10,000× *g*), then 200 μL of supernatant was taken and diluted with 600 μL 80% acetone. The absorbance at 663 nm, 646 nm, and 470 nm were determined using a UV–Vis spectrophotometer (METASH UV5100, Shanghai Metash Instruments Co., Ltd., Shanghai, China). The content of chlorophyll and carotenoids was estimated in mg/g fresh weight (FW).

### 4.4. Analyses of Free Amino Acids in Tea Samples

l-Theanine, l-glutamine, and l-glutamic acid were detected using an ultra-performance liquid chromatography/quadrupole time-of-flight (UPLC−QTOF-MS) system (Waters, Milford, MA, USA) as described previously [25]. Metabolites were separated on a HSS T3 column (1.8 μm, 100 mm × 2.1 mm; Waters, Milford, MA, USA) at 40 °C, with two elution binary gradients at a flow rate of 0.25 mL/min. The mobile phase consisted of water as solvent A containing 0.1% formic acid and acetonitrile as solvent B containing 0.1% formic acid with a gradient program as follows: 0−5 min, 100% A; 5−5.1 min, 100% A to 10% A; 5.1−10 min, 10% A and 90% B; 10−10.1 min, 10% A to 100% A; 10.1−13 min, 100% A. The injection sample volume was 5 μL. The eluted solutions were detected using the following instrument settings in positive ESI mode: capillary voltage was set to at 2 kV with the sampling cone voltage set at 50 V. The source and desolvation temperatures were set at 100 and 300 °C, respectively. The cone and desolvation gas flow rates were set at 50 and 600 L/h, respectively. Data were analyzed by using Mass LynxTM software. The characteristic ions for l-theanine, l-glutamine, and l-glutamic acid are *m*/*z* 175.1083, 147.0770, and 148.0610, respectively.

The method of other free amino acids analyses was referenced from a previous study [73]. The total amount of 100 mg fresh tea samples was extracted with 0.5 mL precooled methanol through ultrasonic extraction for 10 min. For phase separation, 0.7 mL chloroform and 0.2 mL water were added into the centrifuge tubes, and the mixture was centrifuged at 5000× *g* for 10 min. The upper layer was dried using a centrifugal concentrator under vacuum (45 °C, 1400 r/min) and then was dissolved with 5% sulfosalicylic acid (1 mL), standing for 1 h. The solution was centrifuged at 5000× *g* for 10 min and the upper layer was filtered through a 0.22 µm nylon filter membrane. The content of amino acids was determined using an automatic amino acid analyzer (Sykam S-430D, Eresing, Germany). The mobile phase was lithium citrate pH 2.9, pH 4.2, and pH 8.0 with the flowrate of 0.45 mL/min, and the flowrate of the derivatizating reagent was 0.25 mL/min. The instrument parameters were set as follows: automatic injector (5 °C), column temperature (38 °C), post column reaction equipment (130 °C), UV-Vis detection wavelengths (570 nm and 440 nm), and injection volume (50 μL).

### 4.5. Extraction and Analyses of Caffeine, Catechins, and Total Polyphenols in Tea Leaves

The extraction and analysis of caffeine and catechins in tea samples were referenced from a previous method [29]. A total of 100 mg fresh tea samples (fresh weight, finely powdered) was weighted into the 2 mL centrifugal tube and added into 1 mL pre-cooled methanol. After fully vortexing, the mixture was subjected to ultrasonic extraction in ice water for 5 min. The mixture was centrifugated (10,000× *g*, 4 °C) and the supernatants (200 μL) were pipetted into a 10 mL centrifugal tube. Ascorbic acid (1 mM, 600 μL) was added, and the mixture was filtered through the membrane filter (0.22 μm) and subjected to HPLC analysis (Alliance, Waters, Milford, MA, USA). The detection column for the instrument was a ZORBAX Eclipse C18 column (4.6 mm × 150 mm, 5 μm; Agilent, Santa Clara, CA, USA) (40 °C). The mobile phases used were as follows: solvent A was acetonitrile containing 0.1% (*v*/*v*) formic acid and solvent B was Milli-Q water containing 0.1% (*v*/*v*) formic acid. The linear gradient programmer was as follows: 0.0−4.0 min, 12% A−13% A; 4.0−15 min, 13% A; 15.1−22 min, 20% A−40% A; 22.1−25 min, 90% A; 25.1−33 min, 12% A. The injection volume was 10 μL and the detection wavelengths were 280 and 350 nm.

The total polyphenol contents of tea samples were determined using foline-phenol microplate method. Briefly, a total of 100 mg tea samples (fresh weight, finely powdered) were extracted with 70% methanol (2.5 mL), followed by fully vortexing, and then was applied to 70 °C water bath extraction for 10 min. After cooling to room temperature, the extract was centrifuged at 3500 r/min for 10 min, and the supernatant was transferred into a 10 mL centrifuge tube. Then methanol (70%, 2.5 mL) was added into the tea samples residue and the above experimental operations were repeated once again. Then the supernatant was combined and diluted with methanol (70%) to 5 mL and mixed fully. Finally, this tea polyphenol extract was determined at 765 nm using a microplate reader (METASH UV5100, Shanghai Metash Instruments Co., Ltd., Shanghai, China).

### 4.6. Extraction and Analysis of Aroma Compounds in Tea Leaves

The method for extraction and analysis of aroma compounds in tea leaves was based on the previous experimental method with some modifications [65]. A total of 500 mg of tea leaves (fresh weight, finely powdered) was extracted with dichloromethane (2.7 mL), and ethyl decanoate (5 nmol) was used as an internal standard. The mixture was put in a shaker at room temperature overnight and the extraction were dried using anhydrous sodium. The extracts were condensed to approximately 200 μL by blowing dry with a nitrogen blowing concentrator. The extract (1 μL) was then subjected to gas chromatography–mass spectrometry (GC–MS) analysis conforming on a QP2010 SE (Shimadzu Corporation, Kyoto, Japan) equipped with SUPELCOWAX 10 column (30 m × 0.25 mm × 0.25 µm; Supelco Inc., Bellefonte, PA, USA) and GC-MS Solution software (Version 2.72, Shimadzu Corporation, Kyoto, Japan). The working parameters of the instrument were as follow: carrier gas (helium, splitless mode, 1.0 mL/min), injection port (230 °C), GC column temperature program (80 °C for 3 min; then ramp up to 240 °C at 4 °C/min, kept 240 °C for 20 min). Mass spectrometry (Shimadzu Corporation, Kyoto, Japan) was operated in full scan mode (mass range, *m*/*z* 40–200).

### 4.7. Transcript Expression Analysis

The Plant RNA Mini Kit (Waryong, ZH0109) was used to extract total RNA from tea samples, and 1 μg of the total RNA was reversely transcribed into cDNA using the PrimeScript™ RT reagent Kit with gDNA Eraser (Takara, RR047A). Synthesized cDNA (100 ng) was used as the template, 5 μL SYBR Green PCR Super mix (Bio-Rad Laboratories), 2.5 μL primer (10 mM) were mixed as reaction system. The PCR reactions were performed on a Roche LightCycler480 system (Roche Applied Science, Mannheim, Germany). The PCR program included an initial denaturation step at 94 °C for 3 min, followed by 40 cycles of 15 s at 94 °C and 60 s at 60 °C. Each sample was quantified at least in triplicate and normalized using *encoding elongation factor 1-α* (*CsEF1-α*) as an internal control [74]. There were four treatments and each treatment had four replications, amounting for 16 samples. A melt curve was performed at the end of each reaction to verify PCR product specificity. The 2^−Δct^ method was used to calculate relative expression levels.

### 4.8. Statistical Analysis

All statistical data analyses were performed with SPSS 19.0 software. Differences among the different fertilizer treatments were submitted to analysis of one-way analysis of variance (ANOVA) by Tukey’s test and considered statistically significant at *p* < 0.05. All the graphs in this paper were dealt with GraphPad Prism 8.05 and Microsoft Excel 2013 software, respectively.

## 5. Conclusions

In this study, long-term N application increased the expression of key genes for chlorophyll synthesis in tea leaves, promoted the synthesis of chlorophyll, and thus improved the bud density, 100-bud weight, and yield of tea. At the same time, long-term N application not only significantly increased the contents of total free amino acids and the main amino acids (l-theanine, l-glutamic acid, and l-glutamine) in fresh tea leaves, but also decreased the content of catechins, thereby reducing the phenol–ammonia ratio, which is conducive to the formation of tea liquor freshness (Figure 9). Nevertheless, long-term N application led to a significant decrease in the content of benzyl alcohol and 2-phenylethanol, and also down-regulated the expression of *CsNES* and *CsTSB*, which detracted from the formation of floral and fruity aroma compounds in fresh tea leaves. Considering the effects on taste, aroma compounds, growth, and yield, the optimal amount of N fertilizer (225 N kg/hm^2^) balances tea yield and quality. These results provide not only essential information on how N application affects tea quality, but also detailed experimental data for field fertilization of tea plantations.

## Figures and Tables

**Figure 1 metabolites-11-00146-f001:**
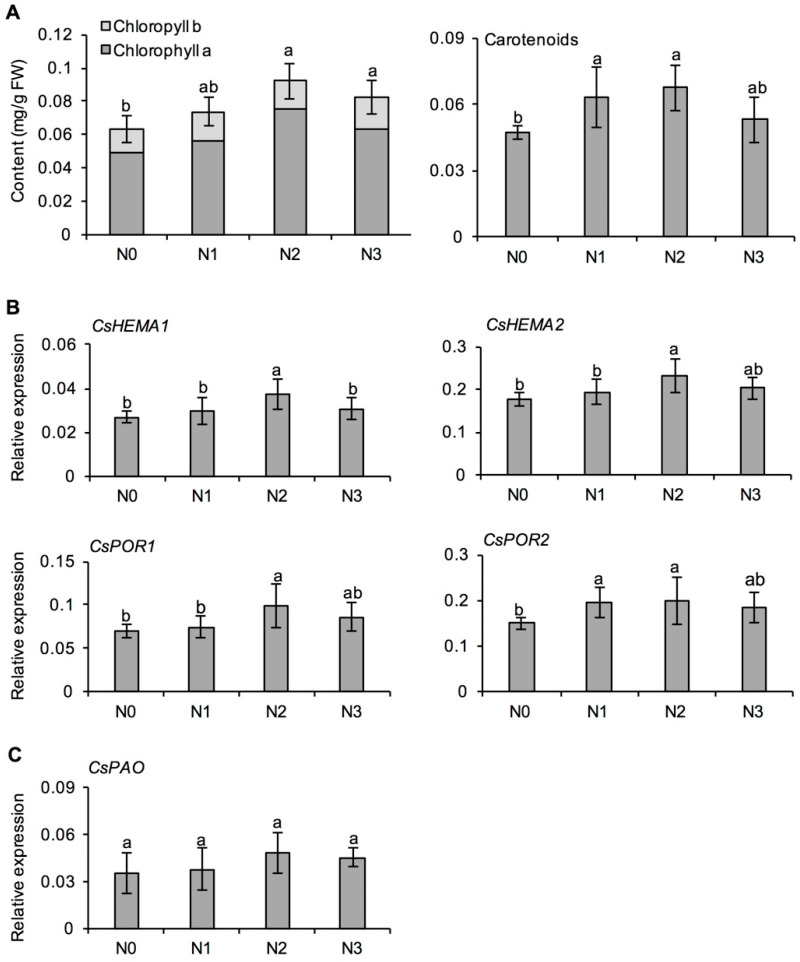
Effects of long-term nitrogen application on the content of chlorophyll and carotenoids (**A**), and expression of key genes for chlorophyll synthesis (**B**), and degradation (**C**) in fresh tea leaves collected in spring. *HEMA*: *glutamyl-tRNA reductase*; *POR*: *protochlorophyllide oxidoreductase*; *PAO*: *pheophorbide a oxygenase*. The nitrogen application treatments indicate N0: 0 N kg/hm^2^; N1: 112.5 N kg/hm^2^; N2: 225 N kg/hm^2^; N3: 450 N kg/hm^2^, respectively. Values followed by different letters (a and b) mean significant at the 5% level among different treatments.

**Figure 2 metabolites-11-00146-f002:**
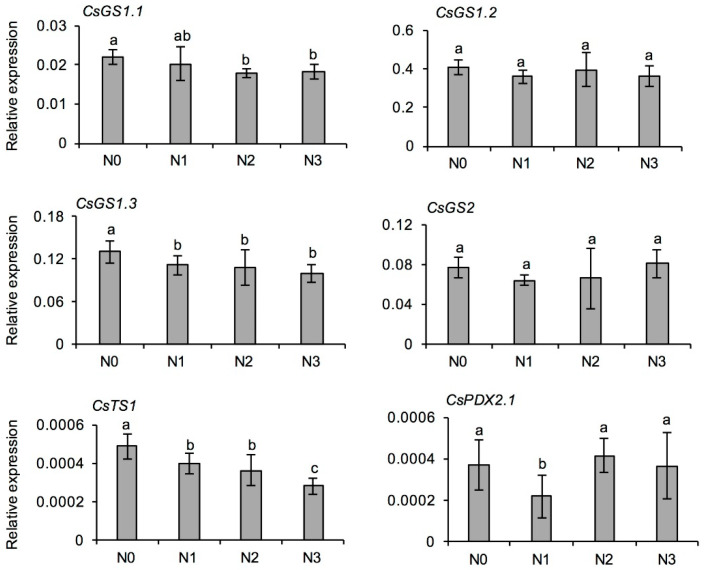
Effects of long-term nitrogen application on the expression of characteristic genes for l-theanine synthesis and degradation in fresh tea leaves collected in spring. *TS*: *l-theanine synthase*; *GS*: *l-glutamine synthase*; *PDX*: *pyridoxal 5′-phosphate synthase subunit*. The nitrogen application treatments indicate N0: 0 N kg/hm^2^; N1: 112.5 N kg/hm^2^; N2: 225 N kg/hm^2^; N3: 450 N kg/hm^2^, respectively. Values followed by different letters (a–c) mean significant at the 5% level among different treatments.

**Figure 3 metabolites-11-00146-f003:**
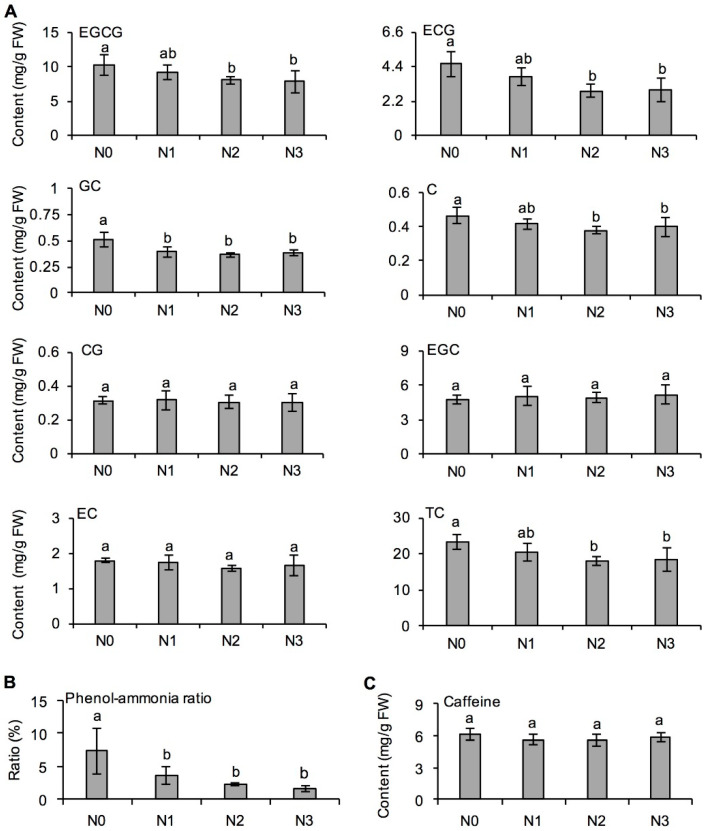
Effects of long-term nitrogen application on catechin content (**A**), phenol–ammonia ratio (**B**), and caffeine content (**C**) in fresh tea leaves collected in spring. EGCG: epigallocatechin gallate; ECG: epicatechin gallate; GC: eallocatechin; C: catechin; CG: catechin gallate; EGC: epigallocatechin; EC: epicatechin; TC: total catechin. The nitrogen application treatments indicate N0: 0 N kg/hm^2^; N1: 112.5 N kg/hm^2^; N2: 225 N kg/hm^2^; N3: 450 N kg/hm^2^, respectively. Values followed by different letters (a and b) mean significant at the 5% level among different treatments.

**Figure 4 metabolites-11-00146-f004:**
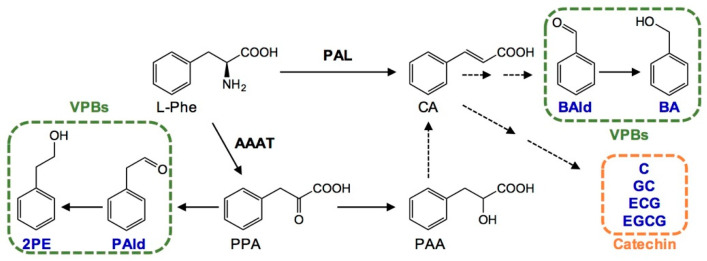
The metabolic pathway of l-phenylalanine in tea leaves. l-Phe: l-phenylalanine; VPBs: volatile phenylpropanoids/benzenoids; BAld: benzaldehyde; BA: benzyl alcohol; CA: *trans*-cinnamic acid; PPA: phenylpyruvic acid; PAA: phenyllactic acid; PAld: phenylacetaldehyde; 2PE: 2-phenylethanol; EGCG: epigallocatechin gallate; ECG: epicatechin gallate; GC: gallocatechin; C: catechin; PAL: phenylalanine lyase; AAAT: aromatic amino acid aminotransferase.

**Figure 5 metabolites-11-00146-f005:**
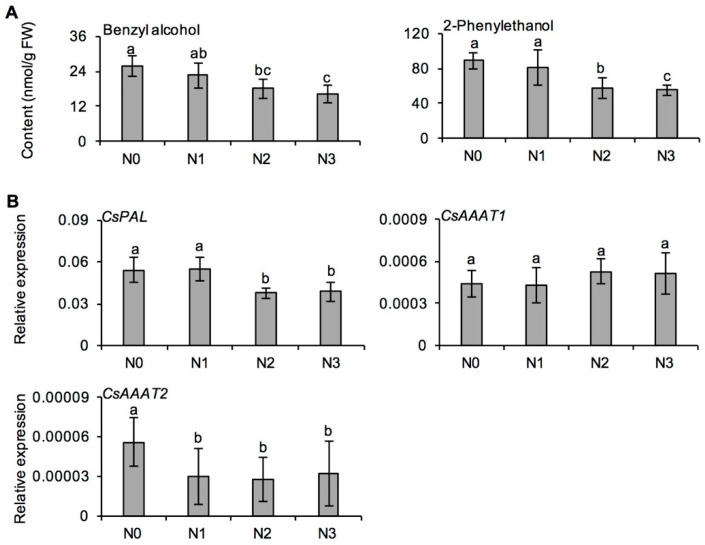
Effects of long-term nitrogen application on the content of aroma compounds derived from l-phenylalanine metabolic pathway (**A**), and expression of *phenylalanine lyase* (*CsPAL*) and *aromatic amino acid aminotransferase 2* (*CsAAAT2*) (**B**) in fresh tea leaves collected in spring. The nitrogen application treatments indicate N0: 0 N kg/hm^2^; N1: 112.5 N kg/hm^2^; N2: 225 N kg/hm^2^; N3: 450 N kg/hm^2^, respectively. Values followed by different letters (a–c) mean significant at the 5% level among different treatments.

**Figure 6 metabolites-11-00146-f006:**
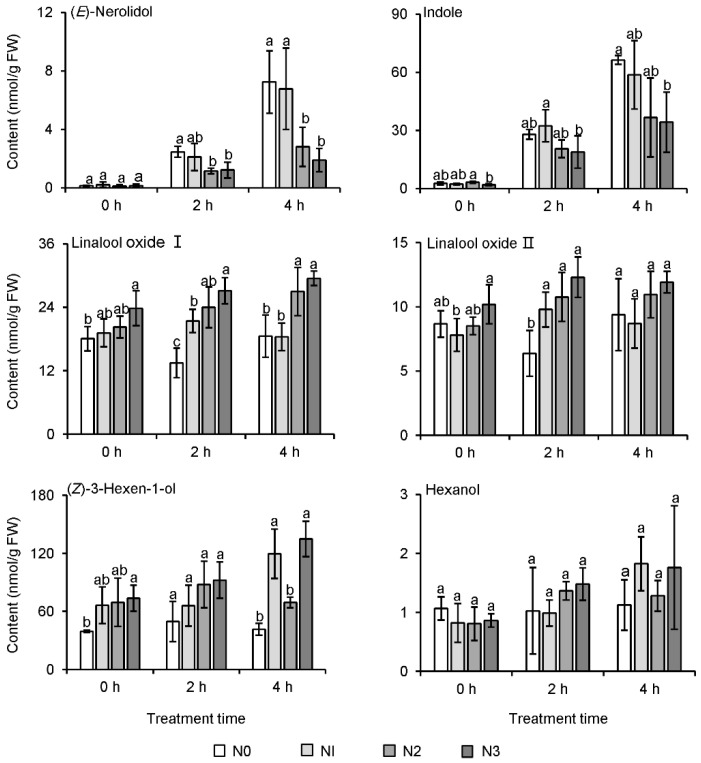
Effect of long-term nitrogen application on the formation of aroma compounds in the withering stage of autumn tea. The nitrogen application treatments indicate N0: 0 N kg/hm^2^; N1: 112.5 N kg/hm^2^; N2: 225 N kg/hm^2^; N3: 450 N kg/hm^2^, respectively. Values followed by different letters (a and b) mean significant at the 5% level at the same treatment time among different treatments.

**Figure 7 metabolites-11-00146-f007:**
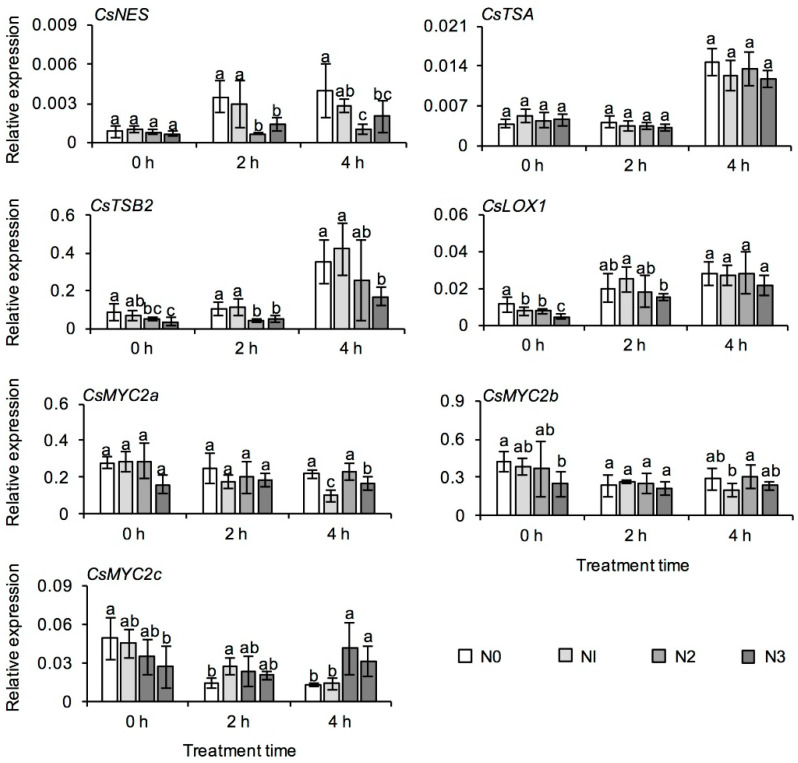
Effect of long-term nitrogen application on the expression of genes involved in aroma compounds formation in the withering stage of autumn tea. *NES*: *(E)-nerolidol synthase*; *TSB*: *tryptophan synthase β-subunit*; *TSA*: *tryptophan synthase α-subunit*; *LOX*: *lipoxygenase*; *MYC*: *myelocytomatosis protein*. The nitrogen application treatments indicate N0: 0 N kg/hm^2^; N1: 112.5 N kg/hm^2^; N2: 225 N kg/hm^2^; N3: 450 N kg/hm^2^, respectively. Values followed by different letters (a–c) mean significant at the 5% level at the same treatment time among different treatments.

**Figure 8 metabolites-11-00146-f008:**
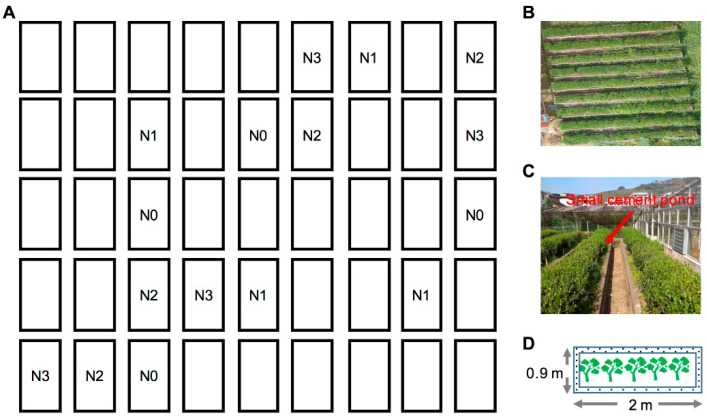
Designation of long-term nitrogen application location filed experiment in a tea garden. (**A**) Each independent small cement pool in the field (blanks are for other treatments); (**B**) Aerial image of the field trials; (**C**) Each plot isolated by concrete ponds; (**D**) The size of each experimental plot.

**Figure 9 metabolites-11-00146-f009:**
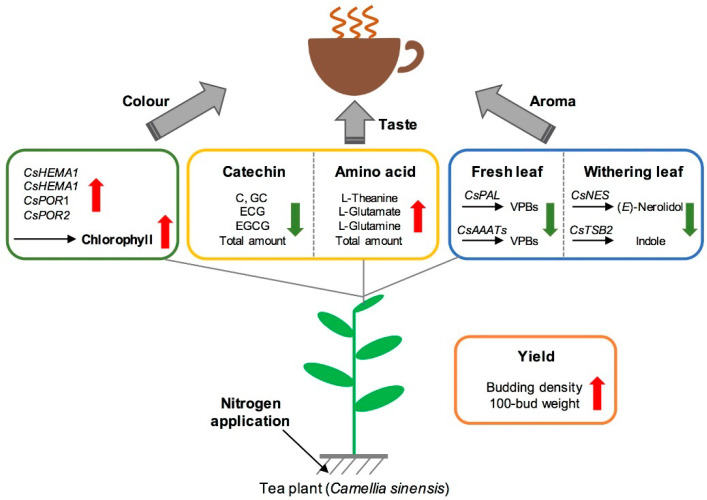
Effects of long-term nitrogen application on tea yield and quality. Red arrows represent the increasing tendency with the increase of nitrogen application, while green arrows represent the opposite tendency. Black arrows represent the nitrogen application. VPBs: volatile phenylpropanoids/benzenoids; EGCG: epigallocatechin gallate; ECG: epicatechin gallate; GC: catechin gallate; C: catechin; *HEMA*: *g**lutamyl-tRNA reductase*; *POR*: *protochlorophyllide oxidoreductase*; *PAL*: *phenylalanine lyase*; *AAAT*: *aromatic amino acid aminotransferase*; *NES*: *(E)-nerolidol synthase*; *TSB*: *tryptophan synthase β-subunit*.

**Table 1 metabolites-11-00146-t001:** Effect of long-term nitrogen application on budding density, 100-bud weight, and yield of spring tea.

Collected Season	Treatment	Budding Density	100-Bud Weight (g)	Yield (kg/hm^2^)
Spring	N0	3681.81 ± 874.85 ^c^	27.07 ± 1.44 ^c^	1.11 ± 0.35 ^c^
	N1	5481.81 ± 1188.47 ^b^	42.93 ± 0.44 ^b^	2.60 ± 0.51 ^b^
	N2	7015.90 ± 924.15 ^a^	63.36 ± 7.60 ^a^	4.90 ± 1.10 ^a^
	N3	7895.45 ± 748.38 ^a^	67.30 ± 3.42 ^a^	5.88 ± 0.63 ^a^

The nitrogen application treatments indicate N0: 0 N kg/hm^2^; N1: 112.5 N kg/hm^2^; N2: 225 N kg/hm^2^; N3: 450 N kg/hm^2^, respectively. Values followed by different letters (^a–c^) mean significant at the 5% level among different treatments.

**Table 2 metabolites-11-00146-t002:** Effect of long-term nitrogen application on budding density, 100-bud weight, and yield of autumn tea.

Collected Season	Treatment	Budding Density	100-Bud Weight (g)	Yield (kg/hm^2^)
Autumn	N0	5031.81 ± 263.00 ^b^	64.38 ± 1.75 ^c^	3.60 ± 0.25 ^c^
	N1	5890.90 ± 714.83 ^a^	72.05 ± 4.10 ^b^	4.69 ± 0.23 ^b^
	N2	6238.63 ± 652.41 ^a^	76.76 ± 5.70 ^b^	5.31 ± 0.68 ^ab^
	N3	6320.45 ± 454.93 ^a^	85.83 ± 3.02 ^a^	6.03 ± 0.53 ^a^

The nitrogen application treatments indicate N0: 0 N kg/hm^2^; N1: 112.5 N kg/hm^2^; N2: 225 N kg/hm^2^; N3: 450 N kg/hm^2^, respectively. Values followed by different letters (^a–c^) mean significant at the 5% level among different treatments.

**Table 3 metabolites-11-00146-t003:** Effects of long-term nitrogen application on the contents of free amino acid components in fresh tea leaves collected in spring.

**Free Amino Acid** **(μg/g FW)**	**N0**	**N1**	**N2**	**N3**
l-Theanine	6812.34 ± 3645.83 ^c^	11,190.05 ± 5058.10 ^bc^	15,328.78 ± 401.43 ^ab^	20,477.66 ± 2475.90 ^a^
l-Glutamic acid	603.84 ± 334.96 ^c^	1026.84 ± 341.09 ^c^	1570.96 ± 116.33 ^b^	2081.24 ± 290.85 ^a^
l-Glutamine	845.10 ± 578.61 ^d^	1729.21 ± 566.34 ^c^	2601.55 ± 265.81 ^b^	3961.21 ± 437.59 ^a^
Serine	220.68 ± 18.62 ^b^	248.12 ± 24.89 ^ab^	295.71 ± 45.67 ^a^	285.02 ± 26.04 ^a^
Threonine	92.54 ± 13.11 ^c^	120.64 ± 14.63 ^b^	113.51 ± 9.20 ^b^	141.04 ± 10.19 ^a^
Alanine	74.57 ± 6.87 ^c^	106.01 ± 12.66 ^b^	132.19 ± 7.55 ^a^	138.90 ± 9.90 ^a^
Valine	50.88 ± 13.25 ^a^	48.63 ± 29.19 ^a^	59.72 ± 32.89 ^a^	29.47 ± 3.09 ^b^
Phosphoserine	61.76 ± 1.99 ^a^	60.19 ± 8.89 ^a^	65.71 ± 4.74 ^a^	57.05 ± 6.22 ^a^
Phosphorylethanolamine	35.43 ± 4.61 ^b^	39.71 ± 1.91 ^ab^	44.50 ± 0.88 ^a^	41.17 ± 2.50 ^a^
Asparagine	20.18 ± 1.56 ^b^	49.87 ± 20.7 ^a^	42.27 ± 7.07 ^a^	47.51 ± 6.73 ^a^
Aspartate	9.60 ± 1.58 ^b^	11.87 ± 1.81 ^ab^	14.13 ± 1.60 ^a^	12.73 ± 1.02 ^a^
Glycine	11.41 ± 1.09 ^a^	12.84 ± 1.62 ^a^	12.76 ± 1.66 ^a^	12.98 ± 1.18 ^a^
Tyrosine	9.66 ± 3.80 ^b^	20.20 ± 7.35 ^a^	22.11 ± 8.58 ^a^	24.42 ± 7.39 ^a^
Phenylalanine	9.85 ± 1.15 ^b^	14.42 ± 2.99 ^ab^	17.27 ± 7.32 ^a^	12.88 ± 0.30 ^ab^
Gamma-aminobutyric acid	17.63 ± 1.10 ^b^	22.53 ± 3.85 ^ab^	29.28 ± 10.32 ^a^	24.62 ± 4.07 ^ab^
Histidine	14.91 ± 3.21 ^c^	24.57 ± 4.02 ^bc^	44.79 ± 19.70 ^a^	32.93 ± 2.05 ^ab^
Cystine	6.39 ± 0.57 ^c^	11.86 ± 7.32 ^a^	8.99 ± 2.56 ^a^	7.97 ± 1.34 ^b^
Leucine	5.37 ± 0.18 ^b^	8.12 ± 1.89 ^ab^	11.03 ± 4.81 ^a^	9.27 ± 2.04 ^ab^
Isoleucine	6.45 ± 1.30 ^c^	9.14 ± 1.87 ^bc^	14.43 ± 6.01 ^a^	12.29 ± 0.96 ^ab^
Tryptophan	27.80 ± 2.90 ^a^	32.49 ± 3.55 ^a^	36.62 ± 12.55 ^a^	27.69 ± 5.88 ^a^
Lysine	6.92 ± 1.18 ^c^	8.69 ± 1.33 ^bc^	13.44 ± 5.02 ^a^	12.33 ± 0.63 ^a^
β-Aminoisobutyric acid	2.20 ± 0.25 ^ab^	1.77 ± 0.19 ^b^	6.38 ± 7.36 ^a^	2.30 ± 0.09 ^ab^
α-Aminobutyric acid	1.25 ± 0.68 ^b^	2.39 ± 2.65 ^a^	1.55 ± 1.32 ^ab^	1.79 ± 1.00 ^ab^
β-Alanine	4.18 ± 0.87 ^b^	4.40 ± 0.77 ^ab^	7.01 ± 3.17 ^a^	4.37 ± 0.15 ^ab^
Taurine	3.44 ± 0.06 ^a^	3.45 ± 0.16 ^a^	3.44 ± 0.14 ^a^	3.50 ± 0.09 ^a^
Total free amino acid content	8949.54 ± 4581.88 ^c^	14,804.82 ± 5983.96 ^bc^	20,499.23 ± 724.62 ^b^	27,464.77 ± 3124.04 ^a^

The nitrogen application treatments indicate N0: 0 N kg/hm^2^; N1: 112.5 N kg/hm^2^; N2: 225 N kg/hm^2^; N3: 450 N kg/hm^2^, respectively. Values followed by different letters (^a–d^) mean significant at the 5% level among different treatments. FW, fresh weight.

## Data Availability

The data presented in this study are available in article and Appendix A
https://www.mdpi.com/2218-1989/11/3/146/s1.

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
