# Peer review of "Effects of Long-Term Nitrogen Fertilization on the Formation of Metabolites Related to Tea Quality in Subtropical China"

_metabolites, 2021, doi:10.3390/metabo11030146_

Round 1

Reviewer 1 Report

In general the paper is well written and relevant references to the previous works in the field are well documented. The work is extensive and interesting  and deserves to be published. I have no comments. 

Reviewer 2 Report

In this work the authors describe their study related to the effects of nitrogen application on the formation of tea quality-related metabolites in a 10-year field experiment. Overall, the article is well written, presenting the results and discussions in detail. However, there are some points that need special attention and corrections before the article can be accepted for publication. Please see my recommendations below.

As a general comment I will point out that some parts of the article are repeated in the manuscript. The authors are kindly asked to read carefully the whole article, trying to condense it and remove the repeated parts.

Please highlight the originally of the article at the end of introduction.

When exactly the samples were collected? After 10 years of applying nitrogen in the mentioned quantities, or year by year during 10 years?

In the note of Table 1 the authors wrote “Values followed by different letters mean significant at the 5% level.” I can see indeed different letters: a, b and c, but I cannot understand what represent each letter. For example, for letter a should be written significant between who and who (between each groups), and so on for each letter. Please clarify.

Lines 102-103 the authors wrote “but there was no significant difference between the 225 kg/hm2 (N2) and 450 kg/hm2 (N3) treatments (p > 0.05)”. When I see Table 1, there looks like all the differences were significant, including N2 and N3. Please explain.

Lines 103-105 again the authors wrote “For autumn tea, the three long term N treatments increased the budding density and 100-bud weight by 17.07–25.61% and 11.91–33.33% (p < 0.05), but there was no significant difference among the three N treatments” The same comment as previous: according with Table 1, all the differences were significant

Lines 109-110 – “but there was no significant difference between the 225 kg/hm2 (N2) and 450 kg/hm2 (N3) treatments”. Same comment. So please clarify the level of significance and the meaning of letters a,b,c.

Figure 2 (line 121) should be Figure 1. However, when discussing Figures 1 or 2 in the text, make reference to each part i.e Figure 1A, Figure 2C, etc. Also, same comment as for Table 1,  define a, b and c in all Figures’ captions.

Lines 225-225 the authors wrote “..... linalool, (E)-nerolidol, and indole, which are characteristic floral or fruity aroma compounds in fresh tea leaves.” I would be very careful with this statement. If linalool has a sweet lavender with a touch of citrus smell and aromas and  nerolidol is providing woody smell/flavor, indole is a very bad smelling compound (poop like), and I am not sure if it can give a pleasant taste.

Lines 244-245 the authors wrote “Compared with leaves at 0 h, the content of indole and (E)-nerolidol in fresh tea leaves increased progressively with withering and was not af- fected by long-term N application to the tea plants”. Connected with my previous comment, related to indole, is obvious that the mentioned component is just a decomposition product, indicating leaf spoilage, as a result of degradation, possibly by microorganisms. Moreover, indole has been detected as a metabolic product of bacteria and perhaps this information should be included in the text.

Lines 246-252 – in my opinion the whole paragraph if highlighting that decomposition products are less occurring in plants that were treated with N fertilizer, rather than having connection with “formation of floral and fruity aroma compounds during the withering process.” Please consider discussing this part accordingly.

Figure 4 A does not have connection with Fig 4B and Fig 4C. Consider presenting Fig 4A as a separate one.

Lines 338-340 please discuss other factors that can affect the normal plant development and trigger secondary metabolites formation ( see DOI: 10.1007/s13762-020-02818-6)

Line 443 449, and 451 Can you consider removing indole from here?

Lines 562-563 Were the leaves crushed before soaking in methanol?

Reviewer 3 Report

  1. Please move the Materials and Methods section before the Results section.
  2. Provide the information on soil types according to World Reference Base for Soil Resources.
  3. Provide the information about an experimental design and a clear justification for the applied rates of P and K fertilizers.
  4. Please remove unnecessary backslashes in the table 1.
  5. Please check the statistical analysis, e.g. there is no statistically significant difference between 1.11±0.35c and 3.60±0.25c (Table 1).

What is the main question addressed by the research?

In the manuscript the authors investigated the effect of different N doses on the tea yield and the formation of its quality-related metabolites.

 Is it relevant and interesting?

Yes

How original is the topic?

Average

Is the paper well written?

Yes

 Is the text clear and easy to read?

Yes

Are the conclusions consistent with the evidence and arguments

presented?

Yes

Do they address the main question posed?

Yes

Round 2

Reviewer 2 Report

The authors addressed all my requirements, and justified reasonable all their answers. The manuscript is considerably improved and clearer. I suggest to be accepted in the present form.